# Tolerance and Effectiveness of Targeted Therapies in Aged Patients with Metastatic Melanoma

**DOI:** 10.3390/cancers13123042

**Published:** 2021-06-18

**Authors:** Ondine Becquart, Bastien Oriano, Stéphane Dalle, Laurent Mortier, Marie Thérèse Leccia, Caroline Dutriaux, Sophie Dalac, Henri Montaudié, Julie De Quatrebarbes, Florence Brunet-Possenti, Philippe Saiag, Thierry Lesimple, Marie Beylot-Barry, Francois Aubin, Pierre-Emmanuel Stoebner, Jean-Philippe Arnault, Brigitte Dreno, Raphael Porcher, Celeste Lebbe, Bernard Guillot

**Affiliations:** 1CHU de Montpellier, Service de Dermatologie, 34295 Montpellier, France; ondine.becquart@gmail.com; 2Hôpital St Louis, APHP, Service de Dermatologie, 75010 Paris, France; bastien.oriano@aphp.fr (B.O.); celeste.lebbe@aphp.fr (C.L.); 3Hôpital Hôtel-Dieu, APHP, Centre d’Épidémiologie Clinique, 75010 Paris, France; raphael.porcher@aphp.fr; 4Service de Dermatologie, Hospices Civils de Lyon, Centre de Recherche en Cancérologie de Lyon, 69002 Lyon, France; stephane.dalle@chu-lyon.fr; 5CHRU Lille, Service de Dermatologie, 59000 Lille, France; laurent.mortiere@chru-lille.fr; 6CHU Grenoble, Service de Dermatologie, 38000 Grenoble, France; mtleccia@chu-grenoble.fr; 7CHU Bordeaux Saint-André, Service de Dermatologie, 33000 Bordeaux, France; caroline.dutriaux@chu-bordeaux.fr (C.D.); marie.beylot-barry@chu-bordeaux.fr (M.B.-B.); 8CHU Dijon, Service de Dermatologie, 21000 Dijon, France; sophie.dalac@chu-dijon.fr; 9CHU Nice, Service de Dermatologie, 06000 Nice, France; montaudie.h@chu-nice.fr; 10CH d’Annecy-Genevois, Service de Dermatologie, 74370 Annecy, France; jdequatrebarbes@ch-annecygenevois.fr; 11Hôpital Bichat, APHP, Paris, Service de Dermatologie, 75018 Paris, France; florence.brunet-possenti@aphp.fr; 12Hôpital Ambroise Pare, APHP, Service de Dermatologie, 92100 Boulogne-Billancourt, France; philippe.saiag@aphp.fr; 13CLCC Eugène Marquis, Service d’Oncologie, 35000 Rennes, France; t.lesimple@rennes.unicancer.fr; 14CHU Jean Mermoz, Service de Dermatologie, 25000 Besançon, France; francois.aubin@univ-fcomte.fr; 15CHU Nîmes, Service de Dermatologie, 30000 Nîmes, France; pierre.STOEBNER@chu-nimes.fr; 16CHU Amiens, Service de Dermatologie, 80000 Amiens, France; arnault.Jean-Philippe@chu-amiens.fr; 17CHU Nantes, Service de Dermatologie, 44000 Nantes, France; brigitte.dreno@atlanmed.fr; 18Departement de Dermatologie, University of Montpellier, 34000 Montpellier, France

**Keywords:** melanoma, targeted therapy, elderly people, side effects

## Abstract

**Simple Summary:**

A majority of melanoma occurs in people over 65 years. BRAF and MEK inhibitors are standard of care for BRAF mutated metastatic melanoma. The aim of the study was to explore tolerability of targeted therapy in a cohort of patients extracted from a biobank. Patients treated by BRAF and/or MEK inhibitors were included in two groups (<65 or >65 years) and analyzed for tolerance and efficacy. The cohort included 353 patients: 231 < 65 years and 122 > 65. A total of 80% had at least one adverse effect mainly skin, general, and gastrointestinal disorders. No statistical difference was observed for severe adverse events, adverse events grades, dose modifications, and interruptions in the two groups. Median overall survival was 20.3 and 16.3 months, respectively. This study shows that tolerance of targeted therapy is as good in older patients as in younger with a similar efficacy. There is no argument against using these treatments in elderly people.

**Abstract:**

Purpose: Melanoma’s incidence is increasing, and elderly people could be significantly impacted since the majority occurs in people over 65 years of age. Combined BRAF and MEK targeted therapies (TT) are current standard regimen for BRAF mutated metastatic melanoma (MM). Except for subgroups of pivotal trials, little data are available for TT in this population. Materials and Methods: Outcomes were explored in real life patients from MelBase, a French multicentric biobank dedicated to the prospective follow-up of unresectable stage III or IV melanoma. Patients treated by BRAF TT and/or MEK TT combined or not, were included from 2013 to 2017 in 2 groups: group 1 ≤ 65-year-old (yo), group 2 > 65 yo, analyzed for tolerance and efficacy. Results: 353 patients were included: 231 in group 1, 122 in group 2. Median follow-up was 12 months (M). Median time of treatment was 6.9 M. A total of 80% had at least one Adverse Effect (AE). Most frequent AE (all grades) were mainly skin and subcutaneous, general, and gastrointestinal disorders. A total of 31% of AE were grade 3–4: 28% in group 1 and 39% in group 2 (*p* = 0.05). No differences were observed in all AE grades proportion, dose modifications, interruptions, and discontinuations. For each group, median overall survival was 20.3 M (CI 95%: 15.5–27.9) and 16.3 M (CI: 14.5–26.9), respectively (*p* = 0.8). Median progression free survival was 7.8 M (6.4–9.9) and 7.7 M (CI: 5.8–11.3) (*p* = 0.4). Objective response rate was 59% and 50% (*p* = 0.6). Conclusion: This study on a large multicentric cohort is the first to assess that TT is well tolerated in elderly BRAF-mutated patients such as in patients younger than 65. Efficacy was similar between groups with outcomes reaching those from pivotal studies. There is thus no argument against using TT in elderly people, although an onco-geriatric opinion is welcome for the most vulnerable.

## 1. Introduction

Melanoma‘s incidence is still increasing annually by 3% [1,2], with more than a quarter diagnosed after 75 years of age [3]. Age-related features make unwieldy the management of cancer [4,5] and conducting specific trials on the elderly is a real challenge [6,7].

Combined BRAF (BRAFi) and MEK inhibitors (MEKi) are now the recommended first line for BRAF mutated Metastatic Melanoma (MM). Vemurafenib with cobimetinib, and dabrafenib with trametinib, respectively, showed response rates of 68 and 67%, overall survival (OS) of 22.3 and 25.1 months, and progression free survival (PFS) of 12.3 and 11 months [8,9,10,11,12,13]. The most common adverse events reported are pyrexia, fatigue, nausea, headache, chills, diarrhea, arthralgia, rash. Severe adverse effects (AE) can affect up to 75% of the treated population and significant impact on the patient’s life quality should not be understated [12,13,14].

The few published data on targeted therapies (TT) used in elderly people have been reviewed [15], and except for pivotal studies subgroups analysis, there is no specific work about their efficacy or safety in this population. Besides, most elderly included in trials were considered to be fit and were not representative. Thus, a French real-life retrospective multicenter study has been conducted to compare the tolerance of TT for BRAF mutated MM’s patients aged over 65 years old with younger patients. Moreover, we analyzed TT effectiveness for those two groups.

## 2. Materials and Methods

### 2.1. Inclusion Criteria

All patients included were analyzed from MELBASE, a French multicentric biobank dedicated to the prospective follow-up for clinical characteristics and outcomes of unresectable stage III and IV melanomas in France since 2013. Protocol was approved by the French ethics committee (CPP Ile-de-France XI, n°12027, 2012), a local ethics committee, and participating institutions. It was registered in the NIH clinical trials database (NCT02828202). Written informed consent was obtained from all patients.

In the present study, from 2 January 2013, until 1 September 2017, patients with V600 BRAF mutated MM treated since at least 3 months with current authorized TT were included: dabrafenib monotherapy, vemurafenib monotherapy, combined vemurafenib and cobimetinib, or dabrafenib and trametinib. Patients followed in Clinical Trials (CT) were excluded except those from the *MEKINIST* trial, which is close to real life conditions (NCT: 02416232). Analyzed population was divided into two groups: group 1: ≤65 yo, group 2: >65 yo. A secondary analysis distinguished group 2a: 65 to 75 yo, and group 2b: >75 yo. Among the whole Melbase cohort, BRAF mutation was observed in 61% of patients. According to age, mutation was observed in 31% of patients older than 65 yo (group 2) and in 27% of patients older than 75 yo (group 2a) (*p* < 0.001).

### 2.2. Study Design

This is a retrospective analysis of a prospective French national database to evaluate tolerance and secondarily to investigate efficacy of the TT in MM for 2 age groups.

### 2.3. Data Collection Methodology

#### 2.3.1. Toxicity Assessment

In Melbase Toxicities were prospectively registered and graded according to the most recent Common Terminology Criteria for Adverse Events (CTC-AE 4.0) from the US National Cancer Institute of 2009 (Table A1). Types of AE were then classified according to the organ-class system used in European medicines agency summaries of treatments characteristics (see details in Appendix A). The other collected data were: TT interruptions, dose modifications, and permanent discontinuations, AE time to onset, and the number of hospitalizations due to toxicity and their duration. Types and grade of AE that led to TT interruption, dose modification, permanent discontinuation, or hospitalization related to adverse events were also collected. In the univariate and multivariate analysis, influencing factors included were: Eastern Cooperative Oncology Group (ECOG) status, Lactate Deshydrogenase (LDH) level, treatment line, stage of disease (M1a, M1b, and M1c according to the 7th edition of American Joint Committee on Cancer Staging), presence of brain metastases, and concurrent radiotherapy.

#### 2.3.2. Efficacy Assessment

Radiological evaluation included brain MRI and total body imaging (PET-scan or CT scan) was performed every three months, and radiological responses were evaluated according RECIST criteria. The Response Rate (RR), progression free survival (PFS), and overall survival (OS) were collected and compared between the 2 groups (a supplementary analysis compared the 3 groups: <65, 65–75, >75 years old). OS was defined as the time interval from the date of the first treatment assumption until death or last date of follow-up, in case of censored observations. PFS was defined as the time interval from the first treatment assumption to the detection of progression or to death from any cause, whichever occurred first. Last visit date was used in case of censored observations. RR was defined as the proportion of patients who presented a complete response or a partial response.

### 2.4. Statistical Analyses

To complete the description, statistical tests were used to compare data between the different subgroups. With a significance level set to 0.05, we applied Chi-square test for qualitative variables and the Kruskal–Wallis test for quantitative variables. Time-to-event analysis were be conducted using the Kaplan–Meier method to estimate survival rate with 95% CI at 6, 12, 18, and 24 months.

To calculate the median follow-up, we chose to use reverse Kaplan-Meier (KM) estimator. The reverse KM survival curve is constructed by reversing “censor” and “event” of the standard KM curve.

To assess the independent effect of each major prognostic factor associated with adverse events onset (at any grade), we used a Cox proportional hazard model. The hypothesis of proportionality of risks over time was verified with Schoenfel residuals.

For the multivariate analysis, we applied a step-by-step method. Main variables were included in the model only if they were associated in bivariate analysis with a *p*-value < 0.25 (via the Wald test). Variables identified according to the backward method: all the predictor variables are entered into the model. The weakest predictor variable is then removed and the regression recalculated. The procedure is repeated until only predictor variables with a threshold <0.005 remain in the model.

### 2.5. Funding Sources

No funding sources were used to conduct this study. MELBASE is a database financed by the National Cancer Institute and a share from the pharmaceutical industry: Roche, BMS, Novartis, MSD.

## 3. Results

### 3.1. Patients

We have described the data as it appears in the database. If the missing data are therefore reported in the table, we applied a multiple imputation for survival data to process them and for our statistical analyzes.

Among the 353 patients included for analysis, 231 were under 65 yo (group 1, 65%), 122 were older, with 72 patients aged between 65 and 75 (group 2a, 20%) and 50 aged over 75 (group 2b, 14%). Median follow up was 12 months (Q1–Q3: 6–17). Median time under treatment was 6.9 months (0.3–58.5). A total of 183 patients (52%) had combination, in which 88 vemurafenib and cobimetinib (25%), 95 dabrafenib and trametinib (27%). Among the 170 patients with monotherapy (48%), 108 had vemurafenib (30%), 55 dabrafenib (15%), 5 cobimetinib (1.4%), and 2 trametinib (0.6%). Proportions of monotherapy in the two groups were similar: 48% (*n* = 110) in group 1, 49% (*n* = 60) in group 2.

Patients and disease characteristics are detailed in Table 1.

### 3.2. Tolerance

Eighty percent of whole population (281 patients/353) had at least one side effect, in which there were 184 for group 1 (80%), 97 for group 2 (80%). There was no significant difference for TT tolerance between age-groups except for the grade ≥ 3 AE (Table 2).

Similar results were found in group 2a (65–75 year old, *n* = 72) and 2b (>75 year-old, *n* = 50) but with more grade ≥3 AE in group 2a (46 vs. 28% of concerned patients). In descending order, most frequent AE were: skin and subcutaneous disorders, general disorders and gastro intestinal disorders (Table 3).

To manage toxicities, dose modifications were necessary for 76 patients (22%) of the whole population, in which there were 46 (20%) and 30 (25%) for groups 1 and 2, respectively (*p* = 0.6). Number of events and proportions of concerned patients are detailed in Table 2.

The most frequent AE leading to dose modification were similar to the most frequent all grades AE.

Grade ≥3 AE led to TT discontinuation or interruption for 77 patients (22%), in which there were 47 (20%) and 30 (25%), respectively.

Twenty seven percent of patients temporary interrupted their treatment, 25% in group 1, 31% in group 2 (*p* = 0.4). The number of events and proportions of concerned patients are detailed in Table 2. The first cause of interruption was the skin and subcutaneous disorders in both populations.

Discontinuations were observed in 20% of all patients and in 18% and 25%, respectively in group 1 and 2 (*p* = 0.6). Number of events and proportions of concerned patients are detailed in Table 2. The first cause of discontinuation was skin and subcutaneous disorders also for the 2 groups, considering all grades AE.

In average, time to onset for the toxicities was 3.9 months for whole population, 4 months for group 1, 3.6 months for group 2 (*p* = 0.7). Figure 1 shows the Kaplan–Meyer curve for appearance of the first AE in the two groups of patients. First AE mostly appeared in the first 3 months. No difference was found between age groups (*p* = 0.8).

#### 3.2.1. Univariate Analyses

In univariate analyses, no association was found between tolerance and treatment line, presence of brain metastases and concomitant radiotherapy (*p* = 0.4, 0.3, and 0.1, respectively). Presence of brain metastases was significantly associated with a higher risk of AE in group 2 (*p* = 0.05). ECOG performance status >0 was linked to a higher risk of poor tolerance of TT in group 1 (HR = 1.4 95% CI (1.1–1.9) *p* = 0.01), and in whole population (HR = 1.3 (1.1–2.0 *p* = 0.02) but not in group 2 HR = 1.5 CI (0.3–2.3) *p* = 0.4.

Advance stage of disease (M1c vs. M1a/b) was inversely related to poor tolerance of TT in the 2 groups and in general population: HR = 0.7 (CI: 0.5–0.9) *p* = 0.04) for group 1, HR = 0.9 (CI 0.2–0.9) *p* = 0.04 in group 2, and HR = 0.7 (CI 0.4–0.9) *p* = 0.04 in whole population.

High LDH level was associated with poor tolerance in every group: HR = 1.7 (CI1.2–2.3) *p* < 0.01 in group 1, HR = 3.3 (CI 1.1–6.4) *p* = 0.05 in group 2. For general population, high LDH presented a significant association with AE HR = 1.8 (CI 1.3–2.8) *p* = 0.03.

#### 3.2.2. Multivariate Analyses

In the whole cohort, moderate advanced disease (stage M1a and M1b) was associated with a poorer tolerance of TT: HR for stage M1c = 0.7 (CI 0.4–0.9) *p* = 0.05. This result was found significant in the two age groups: group 1: HR for stage M1c = 0.6 (CI 0.4–0.9) *p* = 0.04, group 2: HR = 0.8 (CI 0.8–1.0) *p* = 0.05. A high LDH level was also associated with a higher risk of poor tolerance in the whole population: HR = 2.2 (CI 1.1–3.8) *p* = 0.05. This was confirmed in group 1 and 2: HR = 2.4 (CI 1.1–4.6) *p* = 0.05 and HR = 2.6 (CI 1.0–2.2) *p* = 0.05, respectively.

### 3.3. Effectiveness

For each group, median OS was 20.3 M (CI 95%: 15.5–27.9) and 16.3 M (CI: 14.5–26.9), respectively (*p* = 0.8) (Figure 2).

Median PFS were 7.8 M (CI: 6.4–9.9), 7.7 M (CI: 5.8–11.3), respectively in the two groups (*p* = 0.4) (Figure 2).

Similar results were found for groups 2a and 2b for OS or PFS.

Objective RR were 59 and 50%, respectively (*p* = 0.6) (Table 4).

Six months, 12 months, 18 months, and 24 months estimated PFS and OS are detailed in Table 5.

## 4. Discussion

Proportions of elderly in this cohort are higher than in pivotal Clinical Trials (CT) in which the average of over 65 yo was around 30% and over 75 yo subgroups showed a proportion of around 7%. A large pooled analysis of 25 European Organization for Research and Treatment of Cancer (EORTC) trials, confirmed that 9% of over 70 year-old patients were included [16]. Moreover, those included in CT may not be representative of the general status of the elderly. Furthermore, BRAF mutation is less frequent in elderly people as described by Menzies et al. since all patients under 30 had BRAF mutation and only 25% of patients >70 yo had BRAF mutation [17]. In the whole Melbase cohort, 61% of patients had a BRAF mutation. A total of 31% of patients older than 65 yo and 27% of patients older than 75 yo had BRAF mutation. This supports the value of specifically studying elderly people in real life settings [18].

### 4.1. Tolerance

All grades AE in pivotal CT were found between 87 and 98% against 80% in this cohort. They might have been under-collected perhaps due to a less systematic AE record in real-life circumstances.

Grade 3 or 4 AE were found in 31% of the cohort. In reported studies, SAE (defined similarly as grade 3 or 4 AE) were reported at varying degrees (35 to 75%) [9,11,13,14]. It is difficult to correctly compare those rates without a specific study, but AE underreporting cannot be excluded in a real life study.

The only difference between age groups was a slightly significant increase of grade ≥3 AE (28% vs. 39%, *p* < 0.05) for the oldest group (mainly skin reactions, treatment secondary malignancies and renal and urinary disorders). This difference should not be over interpreted since in the complementary analysis, the oldest group 2b (>75 years old) did not show any difference with group 1, rejecting implication of age. The small amount of SAE may have distorted this result.

Comparing to the literature, Larkin et al., in their large cohort with 257 patients aged over 75 yo, showed an increase in grade 3 and 4 AE when compared with people younger (59% of grade 3 vs. 43% and 4% of grade 4 vs. 3%, respectively) and a higher number of AE leading to discontinuation in vemurafenib treated patients [19]. Nevertheless, these outcomes were found exclusively with a BRAFi alone and concerned mostly cutaneous adverse effects that are significantly reduced with MEKi combination that is now recommended. In the CT reported by Robert et al. [11], a large number of patients older than 65 were included, but toxicity was not analysed in different age subgroups.

In a qualitative outlook, profile of tolerance of TT was similar for the two age groups with the skin and subcutaneous reactions at first place. General disorders then gastrointestinal disorders and arthralgia followed. This was consistent with the AE classically found for combined TT [10,11,12,14]. Monotherapy is known to cause higher incidence of cutaneous disorders [20,21,22,23,24,25,26,27] and it could explain the higher proportion of skin AE in this study [28].

To date, there are no known predictive factors of occurrence of AE for TT [29]. Multivariate analyses were conducted in order to highlight predictive factors for poor tolerance. It was found that a higher level of LDH was significantly associated with a higher risk of AE regardless of age. This enzyme has a central function in cellular metabolism and is known to be a poor prognostic factor for MM [30]. To our knowledge its predictive value for TT tolerance has not been highlighted yet.

Alteration of performance status had no impact on TT tolerance in the elderly, while there was no difference of clinical features between groups [31]. This contradicts the assumption that poor ECOG is correlated with poor tolerance as showed by Larkin et al. (ECOG>1 increased rate of AE with vemurafenib alone) [19].

Brain metastases were not correlated to poor tolerance. Except for one real life case reporting a good tolerance and long response to combined TT for a 40 yo patient with brain metastases and ECOG 3 [32]. There is no data supporting a predictive value for TT tolerance of the presence of brain metastases in the literature. In the pivotal trials including brain metastases, outcomes were similar to extra cranial metastases population [30,31,32,33,34,35]

It seemed that early metastatic stage is related to a higher rate of AE regardless of age. Patients at an advanced stage might are less likely to express their side effects.

### 4.2. Effectiveness

In a complementary analysis, the subgroup 2b (over 75) reached a median PFS at 11.4 months. In trial conditions, PFS and OS are respectively reported at 12.3 and 22.3 months for vemurafenib and cobimetinib (8), 11 and 25.1 months for dabrafenib and trametinib [11]. Thus, in terms of effectiveness, in real life conditions, BRAF and MEK TT seemed to show outcomes reaching those in CT.

In the literature, several pivotal studies reported age’s sub-groups analyses for effectiveness and the majority presented no difference or a lack of significance. For monotherapy with vemurafenib compared to dacarbazine, Chapman et al. presented a subgroup of patients over 65 (*n* = 160/337, 23%) that had a HR for OS and PFS similar to the study population, in favor to the BRAF inhibitor. For the group over 65 yo: OS HR was 0.33 (0.16–0.67), PFS HR 0.26 (0.15–0.45), and for whole population OS HR was 0.37 (0.26–0.55) and PFS HR 0.26 (0.20–0.33) [20] For dabrafenib, no study reported age subgroups analysis for efficacy. For trametinib, the HR for PFS was not significant for the subgroup of over 65 yo (*n* = 71/211, 34%): 0.58 (0.29–1.18) (37), but this molecule is now rarely used alone. Subgroups analyses for vemurafenib and cobimetinib in CT showed maintenance of the superiority of combined treatment compared to monotherapy in ages over 65 (*n* = 133/495, 27%)(29). Ascierto et al. confirmed these results in elderly (>65) after a longer follow up with a PFS of 11.2 months, HR: 0·52 (0·34–0·80), and an OS of 24.1 months, HR 0·56 (0·35–0·91) [14]. Concerning dabrafenib and cobimetinib combination, the difference in favor of the targeted therapy was not significant for the 24 to 28% of patients aged over 65 [11,12].

A recent review of the international society of geriatric Oncology agreed with this conclusion [36]. Data for CT of other anti-cancer TT studied in elderly have been reviewed (anti-angiogenic treatment, anti EGFR, M-TOR inhibitors, BRAF and MEK inhibitors, C-KIT inhibitors, HER2 targeted therapies, tyrosine kinase inhibitors, or ALK inhibitors) and showed similar results that for younger patients [15]. A personalized dosage for each patient using pharmacokinetic data could be a new approach for decreasing side effects frequency and severity among elderly and delicate patients [37].

### 4.3. Limitations

Definition of age groups can be discussed. For major health organizations, 65 years is considered as a cut off for “old” patients [38]. In the pivotal studies in melanoma, the threshold for age subgroups was 65 or 75 years. Since mean age in geriatric institutions is 85 yo, 65 years could seem far from the reality threshold of old-age. Nevertheless, we wanted to compare our results to existing subgroup’s data. Dividing the population into three age groups (<65, 65–75, >65) could limit the statistical strength of our results due to the decreased number of patients in each group. Consequently, we analyzed the results for the two groups and the complementary analysis of the three age groups confirmed the absence of statistical difference.

One-year median time to follow up was enough to investigate most tolerance criteria, but short to evaluate effectiveness or long-term AE such as cutaneous carcinogenesis. This population needs to be enlarged with an expanded follow up in order to reinforce the results.

Collection of health-related quality of life is lacking in order to evaluate the real consequences of AE.

## 5. Conclusions

This is the first study evaluating real life outcomes of MAPKinase TT in elderly. Analyses of this large national cohort allow us to consider the absence of difference in tolerance or effectiveness of BRAF and MEK TT in elderly patients compared to patients younger than 65 yo. Faced with the very limited data available in the literature, these results are essential in order to decide eligibility for TT in frail patients.

## Figures and Tables

**Figure 1 cancers-13-03042-f001:**
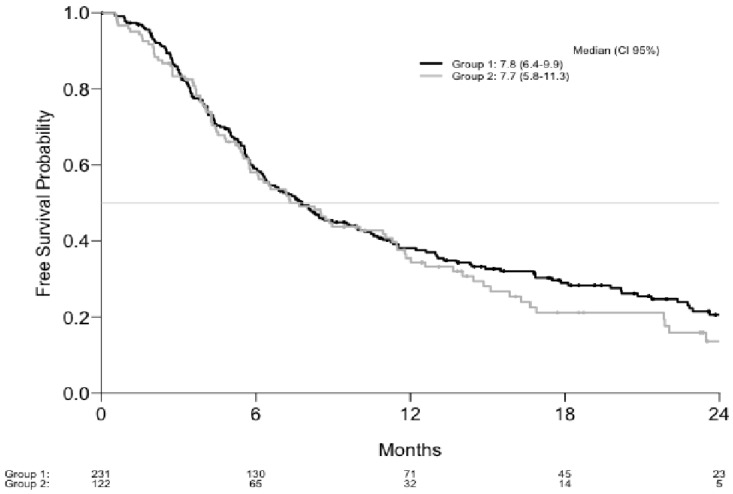
Progression free survival.

**Figure 2 cancers-13-03042-f002:**
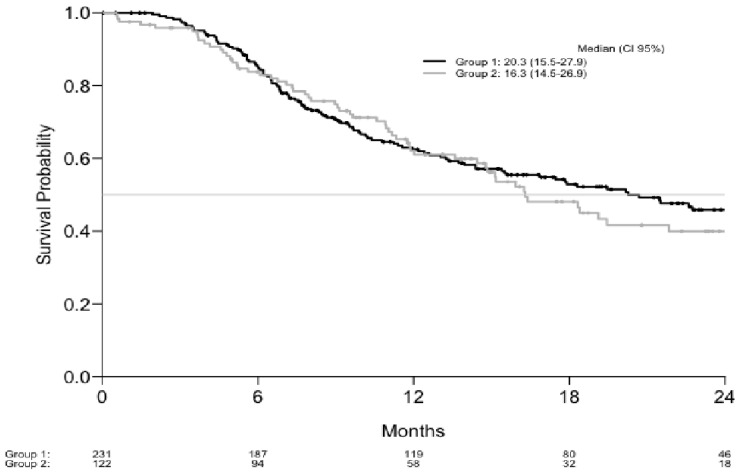
Overall survival.

**Table 1 cancers-13-03042-t001:** Demography, clinical, and treatment characteristics of the cohort.

Demography			
Characteristics	Group 1	Group 2	*p*
Age			
Mean (standard deviation) years	50 (10)	75 (6)	
Median (Q1;Q3) years	51 (43–57)	73 (70–79)	
Gender, N (%)			0.6
Female	96 (42)	50 (41)	
Male	135 (58)	72 (59)	
Clinical characteristics			
**ECOG, N (%)**			0.8
<2	202 (87)	108 (89)	
≥2	29 (13)	14 (11)	
**Stage T**			0.7
1	32 (14)	9 (7)	
2	45 (19)	18 (15)	
3	46 (20)	37 (30)	
4	62 (27)	37 (30)	
Unknown primary	37 (16)	15 (12)	
Unknown Breslow	9 (4)	6 (5)	
**Stage N**			0.6
0	67 (29)	36 (30)	
1	38 (16)	13 (11)	
2	29 (13)	19 (16)	
3	97 (42)	54 (44)	
**Stage M**			0.8
0	21 (9)	10 (8)	
1a	22 (10)	11 (9)	
1b	23 (10)	16 (13)	
1c	165 (71)	85 (70)	
**Stage AJCC**			0.9
III	20 (9)	10 (8)	
IV	211 (91)	112 (92)	
**Nb of affected organs**			0.4
<3	121 (52)	58 (48)	
>3	110 (48)	64 (52)	
**Nb of patients with brain metastases**	61 (26)	30 (25)	0.5
**LDH**			0.2
N missing	28 (12)	13 (11)	
N (%) > x ULN	79 (34)	54 (44)	
N (%) ≤ x ULN	124 (54)	55 (45)	
Treatment characteristics			
**Treatment line**			0.8
First line	185 (80)	102 (84)	
Second line	33 (14)	12 (10)	
Third line	13 (6)	8 (7)	
**Discontinuations, causes**			0,.6
Progression	96 (42)	35 (29)	
Toxicity	41 (18)	34 (28)	
Medical choice	16 (7)	14 (11)	
Patient's choice	7 (3)	2 (2)	
Death	7 (3)	7 (6)	
Unknown	16 (7)	6 (5)	

**Table 2 cancers-13-03042-t002:** Number of events and proportions of patients concerned for all grades adverse effects, grade < 3, grade ≥ 3, dose modification, interruption, discontinuation, hospitalization for BRAF, and MEK inhibitors among the two age groups and whole population.

	Whole Population (*n* = 353)	Group 1 (*n* = 231)	Group 2 (*n* = 122)	*p*
AE (all grade) (*n* event (% of concerned patients))	281 (80)	184 (80)	97 (80)	0.8
AE grade <3 (*n* event (% of concerned patients))	255 (72)	172 (75)	83 (68)	0.5
AE grade ≥3 (*n* event (% of concerned patients))	112 (31)	65 (28)	47 (39)	<0.05
Number of patients who have 0 AE	72 (20)	47 (20)	25 (20)	0.8
Number of patients who have 1 AE	71 (20)	49 (21)	22 (18)	0.7
Number of patients who have 2 AE	46 (13)	28 (12)	18 (15)	0.7
Number of patients who have >2 AE	164 (47)	107 (46)	57 (47)	0.8
Dose modification (*n* event (% of concerned patients))	76 (22)	46 (20)	30 (25)	0.6
Treatment interruption (*n* event (% of concerned patients))	95 (27)	57 (25)	38 (31)	0.4
Treatment discontinuation (*n* event (% of concerned patients))	72 (20)	41 (18)	31 (25)	0.6
Number of hospitalization for toxicity (*n* event (% of concerned patients))	99 (18)	69 (19)	23 (19)	0.8
AE time to onset (months)	3.9	4	3.6	0.7

**Table 3 cancers-13-03042-t003:** All grade adverse effects due to BRAF and MEK inhibitors.

	Adverse Effects Profile	
	Group 1 (n Event (% of Concerned Patients))	Group 2 (n Event (% of Concerned Patients))
**All Grade Adverse Effects**		
Skin and sub cutaneous disorders	265 (50)	141 (57)
General disorders and administration site conditions	166 (40)	87 (33)
Gastrointestinal disorders	102 (27)	64 (30)
Musculoskeletal and systemic disorders	67 (20)	21 (15)
Investigations	55 (11)	19 (8)
Nervous system disorders	28 (10)	4 (2)
Hematologic and lymphatic disorders	23 (7)	16 (7)
Ocular manifestations	21 (9)	18 (11)
Renal and urinary disorders	17 (6)	25 (13)
Non-precised malignant and benign tumors	9 (3)	20 (7)
Vascular Disorders	17 (6)	14 (9)

**Table 4 cancers-13-03042-t004:** Best response rates.

	Group 1 *n* = 231	Group 2 *n* = 72	*p*
Best response N (%)			
Complete Response (CR)	45(19)	20 (16)	0.8
Partial Response (PR)	91 (39)	41 (34)	0.7
Stable Disease (SD)	55 (24)	33 (27)	0.8
Progressive Disease (PD)	40 (17)	28 (23)	0.4
Objective response (CR+ PR) N(%)	136 (59)	61 (50)	0.6
Disease control (R + PR + SD) N (%)	191 (83)	94 (77)	0.7

**Table 5 cancers-13-03042-t005:** Estimated PFS and OS.

Progression Free Survival Estimations			Overall Survival Estimations		
Months	Estimation (%)	CI 95%	Months	Estimation (%)	CI 95%
Group 1 (%)			Group 1 (%)		
6	59.1	53.0–65.9	6	85.2	80.7–90.0
12	38.1	32.0–45.2	12	62.4	56.2–69.4
18	29.0	23.3–36.1	18	52.9	46.3–60.3
24	20.6	15.3–27.8	24	45.9	39.0–53.9
Group 2 (%)			Group 2 (%)		
6	58.1	49.7–67.8	6	83.8	77.4–90.7
12	35.4	27.5–45.7	12	62.1	53.6–71.9
18	21.1	14.1–31.6	18	48.1	38.9–59.5
24	13.6	7.5–24.8	24	39.9	30.5–52.2
Total (%)			Total (%)		
6	58.7	53.7–64.2	6	84.7	81.0–88.6
12	37.2	32.3–42.9	12	62.3	57.2–67.9
18	26.6	21.9–32.2	18	51.5	46.1–57.6
24	18.5	14.1–24.2	24	44.2	38.5–50.7

## Data Availability

Data is contained within the article.

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
