# Peer review of "Tolerance and Effectiveness of Targeted Therapies in Aged Patients with Metastatic Melanoma"

_cancers, 2021, doi:10.3390/cancers13123042_

Round 1

Reviewer 1 Report

My only comments should be considered minor ones.

  1. Although they discuss the paper by Larkin et al., there have been other clinical trials that included a large cohort of elderly patients, for example Robert et al. in their reference 18.  I think the paper would benefit from a fuller acknowledgement of previous observations of TT in elderly patients on clinical trials.
  2. The reasons for discontinuation (Table 1) indicate that elderly patients were less likely to discontinue due to progression but more likely to discontinue for toxicity or medical choice.  Were these differences statistically significant?  If so, this might be an interesting aspect to discuss further.

Author Response

Thank you for the comments of the two reviewers allowing an improvement of our paper.

Concerning the first reviewer, Robert et al's article has been more widely exploited and discussed and causes of treatment discontinuation was statisticaly analysed. No diffrence could be observe among the causes of discontinuation in the two groups.

Reviewer 2 Report

The clinical utility of treatment with BRAF inhibitors (vemurafenib and dabrafenib) alone or, currently, in combination with MEK inhibitors (cobimetinb and trametinib), is limited to melanomas carrying a BRAF kinase mutation. Several studies show that the frequency of occurrence of BRAF mutations correlates inversely with age. In an Australian cohort (A.M. Menzies et al. Distinguishing clinicopathologic features of patients with V600E and V600K BRAF-mutant metastatic melanoma. Clin Cancer Res., 18 (2012), pp. 3242-3249) of more than 300 patients with metastatic melanoma, all patients under 30 years of age had a BRAF mutation, while only 25% of those over 70 years of age had a BRAF mutation. Interestingly, in the elderly population, the proportion of the most frequent BRAF mutation, V600E, decreases, while other less frequent BRAF mutations, such as BRAF, V600K, increase in frequency. Although the underrepresentation in clinical trials of elderly patients is a global problem in oncology, in target therapy trials this decrease in BRAF mutation frequency surely also contributes to this. Although in this series the proportion is higher due to is near to real life setting, only 14% were aged over 75. It would be interesting for the reader to know what percentage of the total sample of patients older than 75 years had BRAF mutated metastatic melanoma. 

Author Response

Thank you for the comments of the two reviewers allowing an improvement of our paper.

Concerning comments of reviewer 2, the percentage of BRAF mutated patient in the whole Melbase cohort and in each group of age has been done. The results on a less important percentage of mutation in elderly patients is discussed on the light of the apper of Menziez et al.